# CLEF: Contrastive Learning of Equivariant Features in CT Images

Ilya Kuleshov[1], Mikhail Goncharov[2], and Vera Soboleva[2]

[1] Moscow Institute of Physics and Technology, Russia
[2] Skoltech Institute, Russia

**Abstract.** This work focuses on developing a self-supervised method of pretraining on biomedical images. The pretrained models are then fine-tuned on a small labelled dataset. We show, that using contrastive learning along with an equivariance loss and a loss, designed by us to maximise the features' information, we manage to improve quality in comparison to a fully-supervised baseline. Our method of pretraining achieves an average dice score of 0.86, reducing the baseline error by 20%.

**Keywords:** Self-supervised learning · Biomedical image segmentation · Contrastive learning.

## 1 Introduction

The biomedical datasets with labelled image are very limited due to high complexity of the process of labelling these images. This gives way to the idea of self-supervised pretraining on unlabelled images. We propose to force the learned features, corresponding to a voxel in the human body, to reflect it's anatomical location. This would guarantee high quality on the task of organ segmentation, which is the task at hand. To achieve said quality, we propose a compound, three-part loss, which would force the features to behave similarly to such a general anatomical system of coordinates.

## 2 Method

### 2.1 Preprocessing

In this section we will describe our preprocessing strategy. The transformations are as follows.

1. A mask of all voxels with intensities, greater than $-500$HU is generated.
2. The image is cropped to the smallest box, containing the mask, generated in the previous step.
3. The image is resized (via interpolation) to shape $(192, 192, 192)$.
4. If necessary, the axes are flipped, so that the resulting image has canonical orientation.
5. The intensities are clipped to the window $(-200, 300)$, the HU interval in which most soft tissues reside.
6. Finally, the intensities are scaled to the range $(-1, 1)$

## 2.2   Proposed Method

Our key contribution lies in the self-supervised pretraining on the large unlabeled part of our dataset. The pretraining method utilises three losses, each accomplishing a different objective. We use the notation $f_{enc}$ to depict the encoding part of our network, and $X$ for the preprocessed input image.

**Pretraining: Decoding Loss** Firstly, we want to ensure that the encoded features contain the information about the intensity of the original voxel. For that purpose we use the decoding head: two 1x1 convolutions, applied to the output features. We minimise the mean squared error between the output of our decoding head $f_{dec}$ and our original image $X$:

$$L_{dec} = MSE(f_{dec}(f_{enc}(X)), X) = \frac{1}{N} \sum_{i,j,k} (f_{dec}(f_{enc}(X))[i,j,k] - X[i,j,k])^2$$

**Pretraining: Discriminativeness loss** We also aim to be able to guess the location of a voxel by it's features. Thus, we add a loss which forces the model to predict distinct features, such that voxels, located far apart from each other, have different representations. After getting the features of each voxel, we randomly sample a small subset of anchor voxels $I_A$ and a large subset of leaf voxels $I_L$ and compute the pairwise distance in feature space (negative inner product) between the anchors and the leaves $D$:

$$D[i_A, i_L] = -\langle X[i_A] \cdot X[i_L] \rangle; i_A \in I_A, i_L \in I_L$$

Next, for each anchor $i_A$, we compute the indices of the voxels which are far enough from it, $F(i_A)$. Typically, we considered voxels to be far enough from each other if the euclidean distance between them was $\geq 10$mm, so $F(i_A)$ can be computed as the following (we use $\|\cdot\|_{mm}$ to denote the physical distance in milimeters between voxels):

$$F(i_A) = \{i \in I_L : \|i - i_A\|_{mm} \geq 10\}$$

Finally, we apply the following activation function:

$$L_d = \sum_{i_A \in I_A} \sum_{i_L \in F(i_A)} \text{relu}(M - D[i_A, i_L]) \tag{1}$$

Here, $M$, margin, is a hyperpameter, we set it to $-0.9$ in our experiments. Such a loss forces the model to cluster features of close voxels together. The higher $M$ is, the less is the amount of possible feature vectors, located from each other at a distance, greater than $M$. For example, if our feature space is a 3D sphere and $M = 0$, then there are at most 8 vectors $\{v_i\}_{i=1}^8$, which all satisfy the inequality $(v_i, v_j) \geq M$ (all of such vectors are either perpendicular to each other, or facing in opposite directions from each other).

**Pretraining: Equivariance loss** Finally, we also demand the equivariance property from our model. Although this accomplishes more or less the same task as our decoder, we decided to include this loss, as it is easily learned by the model. One of many strenghts of convolutional networks is that the object's representation is independent of it's location in the image, which implies equivariance, at least in relation to shifts. In addition to shifts, we also train our neural network to be equivariant in relation to zooming and rotation. The loss is the negative inner product between a randomly rotated, scaled and shifted representation of the original image, and the representation of the transformed image. In other words, if we denote $T$ as our random affine transformation, $X$ as our image, and $f$ as our neural network, the loss is as follows:

$$L_e = -\langle T(f(X)) \cdot f(T(X)) \rangle \tag{2}$$

**Finetuning** We fine-tune with a compound loss function, which is the summation between Dice loss and cross entropy loss. This kind of loss function has proven to be effective in biomedical image segmenation [4].

**Architecture** We use a two-part architecture, consisting of a large, feature-extracting backbone and a small head. The backbone is a simple 3D U-Net. The head consists of two 1x1 convolutions, which we apply to the output features of the backbone. Figure 1 illustrates the applied 3D U-Net [6]. The number of channels the head outputs determines the dimensionality of our feature space in the case of pretraining, whereas in the case of fine-tuning it must be equal to the number of classes, thus when transferring weights from the pretraining model to the fine-tuning model, we transfer only the weights of the backbone U-Net, while the head is re-initialized with fresh random weights.

### 2.3   Post-processing

The fine-tuning predictions are passed through a sigmoid function, thus scaling to a $(0, 1)$ range. Then, for each voxel: if all of the resulting logits are less than .5, we deduce that voxel to be outside of any organs, which interest us in this task. Otherwise, that voxel is labelled with the index of the largest logit value in it's predicted vector.

## 3   Experiments

### 3.1   Dataset and evaluation measures

The FLARE2022 dataset is curated from more than 20 medical groups under the license permission, including MSD [7], KiTS [2,3], AbdomenCT-1K [5], and TCIA [1]. The training set includes 50 labelled CT scans with pancreas disease and 2000 unlabelled CT scans with liver, kidney, spleen, or pancreas diseases. The validation set includes 50 CT scans with liver, kidney, spleen, or pancreas

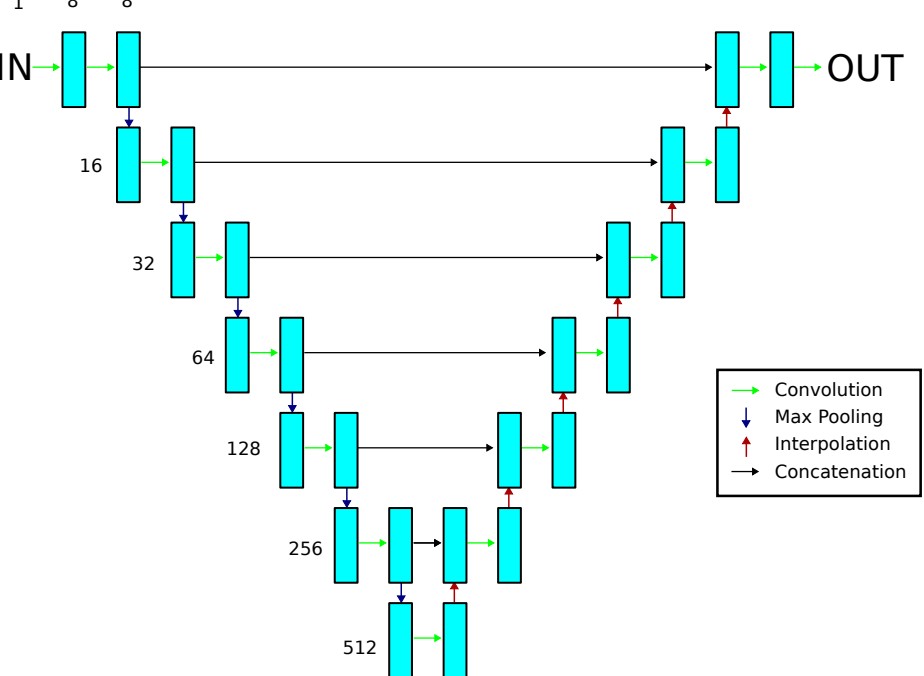

**Fig. 1.** Network architecture

diseases. The testing set includes 200 CT scans where 100 cases has liver, kidney, spleen, or pancreas diseases and the other 100 cases has uterine corpus endometrial, urothelial bladder, stomach, sarcomas, or ovarian diseases. All the CT scans only have image information and the center information is not available.

The evaluation measures consist of two accuracy measures: Dice Similarity Coefficient (DSC) and Normalized Surface Dice (NSD), and three running efficiency measures: running time, area under GPU memory-time curve, and area under CPU utilization-time curve. All measures will be used to compute the ranking. Moreover, the GPU memory consumption has a 2 GB tolerance.

### 3.2 Implementation details

**Environment settings** The development environments and requirements are presented in Table 1.

**Table 1.** Development environments and requirements.

| | |
|---|---|
| Windows/Ubuntu version | Ubuntu 18.04.5 LTS |
| CPU | Intel(R) Core(TM) i9-7900X CPU@3.30GHz |
| RAM | 16×4GB; 2.67MT/s |
| GPU (number and type) | Four NVIDIA V100 16G |
| CUDA version | 11.0 |
| Programming language | Python 3.9 |
| Deep learning framework | Pytorch (Torch 1.10, torchvision 0.2.2) |
| Specific dependencies | |
| (Optional) Link to code | |

**Training protocols** We used no data augmentation, passing the whole $192 \times 192 \times 192$ image to the model without random cropping. The equivariance loss requires a random affine transform. We randomly sampled two of three axes, and applied a random transform in the corresponding plane. For the transform, we used a combination of a random scale in the range $(1, 1.5)$, a random shift in the range $(-0.1, 0.1)$ on each of the selected axes, and a random rotation in the range $(-30°, 30°)$ in the selected plane.

When fine-tuning, we randomly divide the 50 labeled samples into 5 folds. One of the five resulting subsets is used for validation the others are used for training. We use early stopping to stop our model from overfitting on the small dataset of labelled samples: when the loss on the validation set does not decrease for three epochs, the training stops. This results in the training being cut short after 18-22 epochs, depending on the chosen fold.

**Table 2.** Training protocols.

| Network initialization | default PyTorch initialization |
|---|---|
| Batch size | 1 |
| Total epochs | 50 |
| Optimizer | Adam |
| Initial learning rate (lr) | 0.0004 |
| Training time | 15.5 hours |
| Number of model parameters | 33.0M[3] |

**Table 3.** Training protocols for the fine-tuning model (if using two-stage framework).

| Network initialization | default PyTorch initialization |
|---|---|
| Batch size | 1 |
| Total epochs | 50/Early stopping |
| Optimizer | Adam |
| Initial learning rate (lr) | 0.001 |
| Training time | 1-2 hours |
| Number of model parameters | 33.0M[4] |

## 4    Results and discussion

### 4.1    Evaluation on the validation set

The self-supervised pretraining on unlabelled cases provided an improvement in comparison to a model, trained in a supervised fashion. The model works very well on clearly visible organs, which can be visually separated from their surroundings. Such organs include the liver, kidneys, the aorta and some others. Less visible organs, such as the duodenum (see Figure 2), the pancreas (see Figure 3), the left adrenal gland proved to be more complicated for our method, which is to be expected. But, according to Table 4, there are some isolated cases, in which highly-visible organs have lower quality. This is due to anomalies, as can be seen on Figure 4. The average dice score after pretraining on the validation set is 0.87, which is a rather big improvement from the baseline average dice score (0.84).

**Table 4.** Segmentation results

|  | Liver | RK | Spleen | Pancreas | Aorta | IVC | RAG | LAG | GB | Esophagus | Stomach | Duodenum | LK |
|---|---|---|---|---|---|---|---|---|---|---|---|---|---|
| 0 | 0.98 | 0.96 | 0.97 | 0.80 | 0.93 | 0.90 | 0.88 | 0.86 | 0.86 | 0.82 | 0.95 | 0.68 | 0.96 |
| 1 | 0.98 | 0.97 | 0.98 | 0.78 | 0.96 | 0.92 | 0.74 | 0.75 | 0.93 | 0.85 | 0.95 | 0.71 | 0.97 |
| 2 | 0.97 | 0.97 | 0.98 | 0.85 | 0.95 | 0.93 | 0.93 | 0.88 | 0.69 | 0.88 | 0.93 | 0.81 | 0.96 |
| 3 | 0.98 | 0.97 | 0.96 | 0.75 | 0.96 | 0.85 | 0.91 | 0.90 | 0.91 | 0.80 | 0.84 | 0.63 | 0.98 |
| 4 | 0.98 | 0.96 | 0.98 | 0.78 | 0.95 | 0.89 | 0.89 | 0.81 | 0.93 | 0.77 | 0.93 | 0.79 | 0.89 |
| 5 | 0.96 | 0.98 | 0.97 | 0.80 | 0.94 | 0.80 | 0.50 | 0.78 | 0.80 | 0.87 | 0.88 | 0.84 | 0.97 |
| 6 | 0.97 | 0.51 | 0.98 | 0.80 | 0.96 | 0.87 | 0.88 | 0.78 | 0.94 | 0.78 | 0.94 | 0.80 | 0.97 |
| 7 | 0.98 | 0.96 | 0.97 | 0.52 | 0.92 | 0.90 | 0.75 | 0.87 | 0.90 | 0.89 | 0.95 | 0.57 | 0.89 |
| 8 | 0.98 | 0.97 | 0.98 | 0.79 | 0.94 | 0.91 | 0.83 | 0.84 | 0.93 | 0.81 | 0.91 | 0.78 | 0.98 |
| 9 | 0.97 | 0.93 | 0.98 | 0.59 | 0.95 | 0.92 | 0.83 | 0.63 | 0.91 | 0.88 | 0.92 | 0.59 | 0.40 |
| mean | 0.98 | 0.92 | 0.97 | 0.75 | 0.95 | 0.89 | 0.81 | 0.81 | 0.88 | 0.83 | 0.92 | 0.72 | 0.90 |

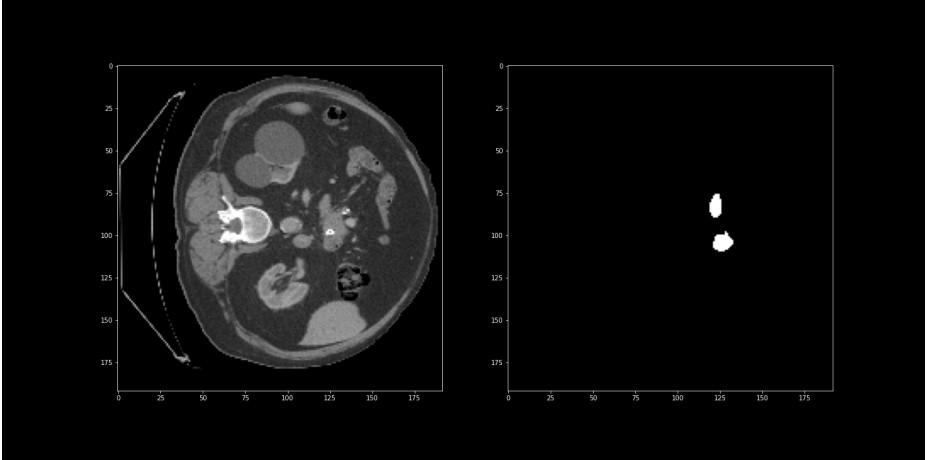

**Fig. 2.** A slice and the corresponding duodenum mask.

### 4.2   Validation results

Here are some segmentation maps on the validation set: subsection 4.2, Figure 4.2, Figure 4.2, Figure 4.2, Figure 4.2. As instructed, all segmentation maps are presented in a window, centered at 40HU, with a width of 400HU.

### 4.3   Results on final testing set

The results on the testing set can be seen in Table 5, they are worse than the results on the validation set, probably due to a slightly skewed distribution.

### 4.4   Limitation and future work

Our method shows great promise. Nevertheless, it can be modified in many ways. These include, but are not limited to the following.

– Adding preprocessing. This could help with artifacts that are visible on some predicted masks (see Figure 10).
– This method could be improved by predicting several neighbouring image voxels by the MLP head, instead of a single voxel. This should help store information on the surrounding voxels in pretrained features, forcing our model to learn better features.

## 5   Conclusion

Contrastive self-supervised pretraining helps improve quality of the resulting fine-tuned network. This means, that our pretraining method forces the backbone to learn informative features, which, at least in part, carry information on human organs.

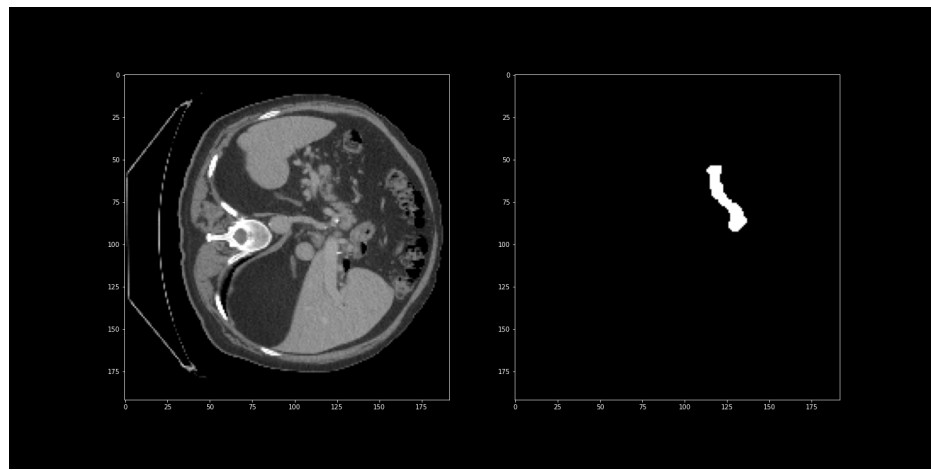

**Fig. 3.** A slice and the corresponding pancreas mask

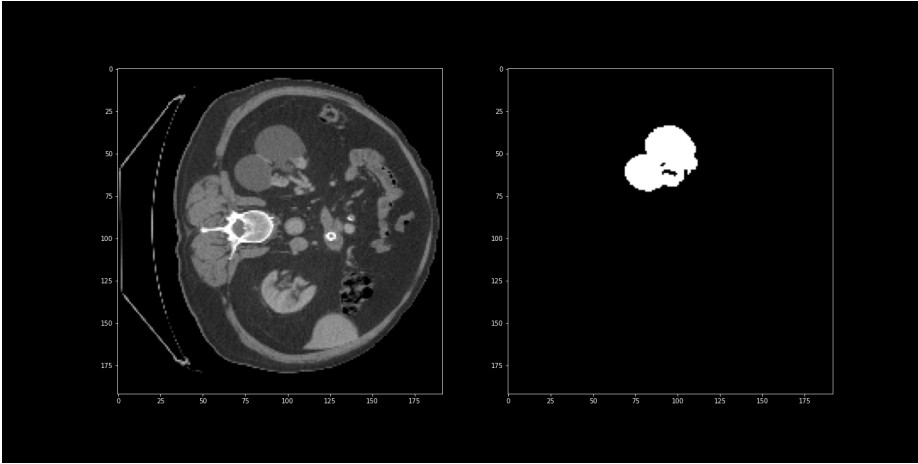

**Fig. 4.** An anomaly in the left kidney, image FLARE22_Tr_0045

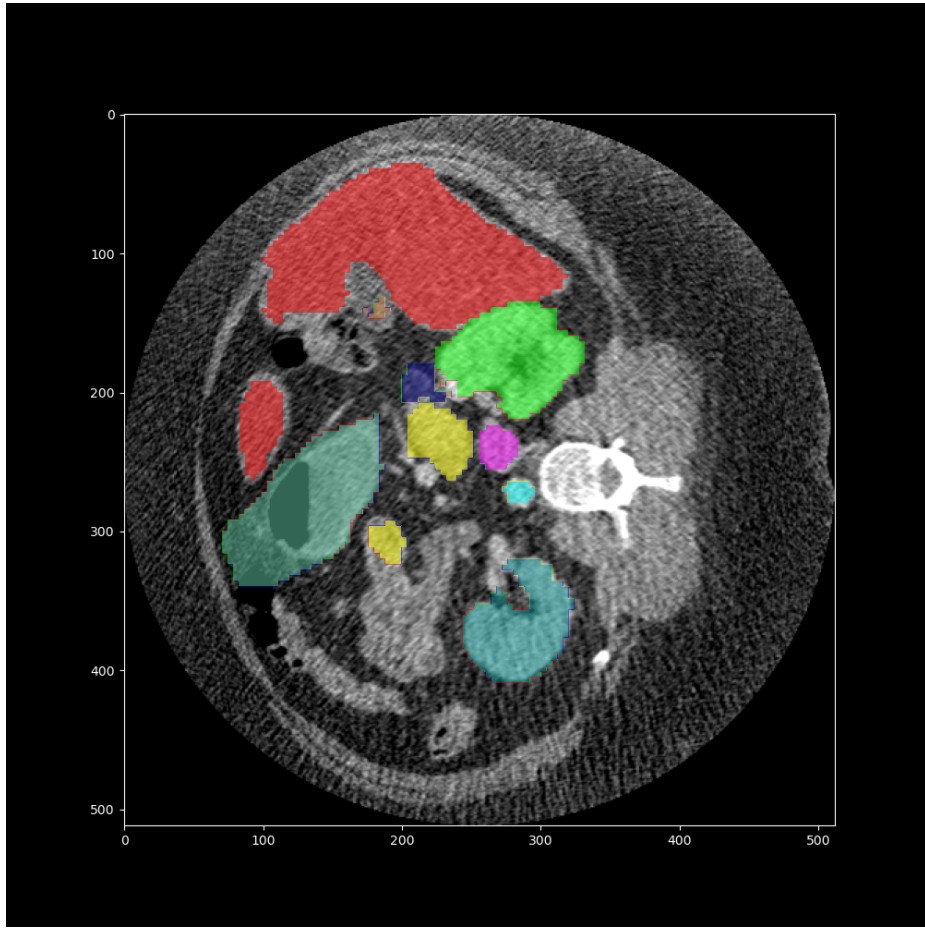

**Fig. 5.** Segmentation results on case 1

**Acknowledgements** The authors of this paper declare that the segmentation method they implemented for participation in the FLARE 2022 challenge has not used any pre-trained models nor additional datasets other than those provided by the organizers. The proposed solution is fully automatic without any manual intervention.

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

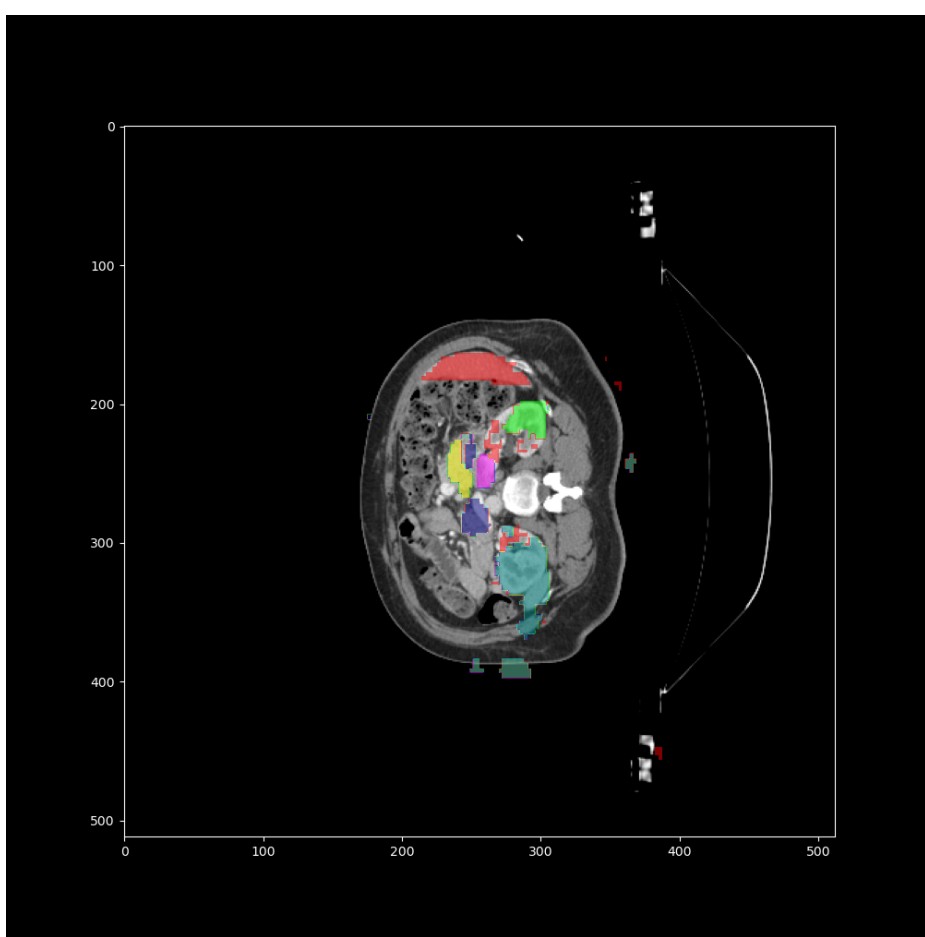

**Fig. 8.** Segmentation results on case 30

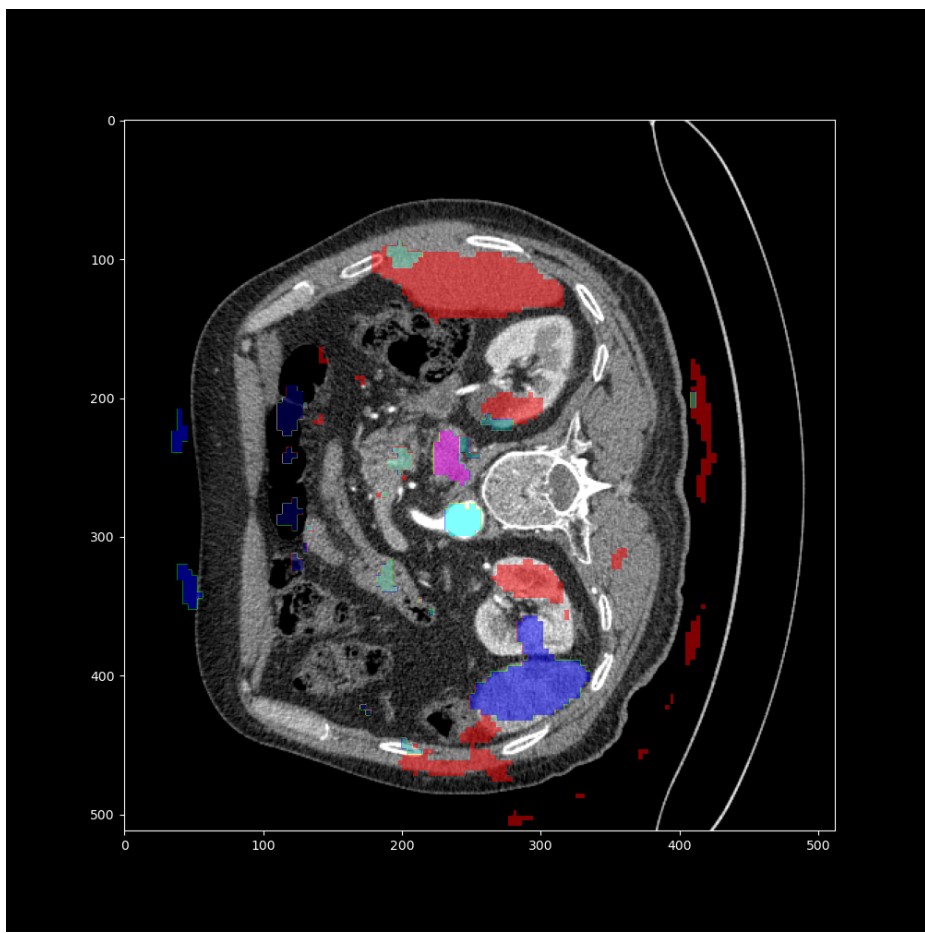

**Fig. 9.** Segmentation results on case 50

**Table 5.** Testing results

| Name | Mean | STD |
|---|---|---|
| Liver_DSC | 0.89 | 0.09 |
| RK_DSC | 0.84 | 0.21 |
| Spleen_DSC | 0.85 | 0.19 |
| Pancreas_DSC | 0.53 | 0.18 |
| Aorta_DSC | 0.81 | 0.12 |
| IVC_DSC | 0.73 | 0.12 |
| RAG_DSC | 0.02 | 0.04 |
| LAG_DSC | 0.01 | 0.1 |
| Gallbladder_DSC | 0.14 | 0.17 |
| Esophagus_DSC | 0.53 | 0.21 |
| Stomach_DSC | 0.66 | 0.2 |
| Duodenum_DSC | 0.47 | 0.19 |
| LK_DSC | 0.81 | 0.22 |
| Liver_NSD | 0.78 | 0.17 |
| RK_NSD | 0.81 | 0.23 |
| Spleen_NSD | 0.83 | 0.22 |
| Pancreas_NSD | 0.61 | 0.17 |
| Aorta_NSD | 0.77 | 0.16 |
| IVC_NSD | 0.65 | 0.13 |
| RAG_NSD | 0.09 | 0.09 |
| LAG_NSD | 0.01 | 0.1 |
| Gallbladder_NSD | 0.16 | 0.17 |
| Esophagus_NSD | 0.65 | 0.22 |
| Stomach_NSD | 0.59 | 0.21 |
| Duodenum_NSD | 0.73 | 0.19 |
| LK_NSD | 0.76 | 0.24 |

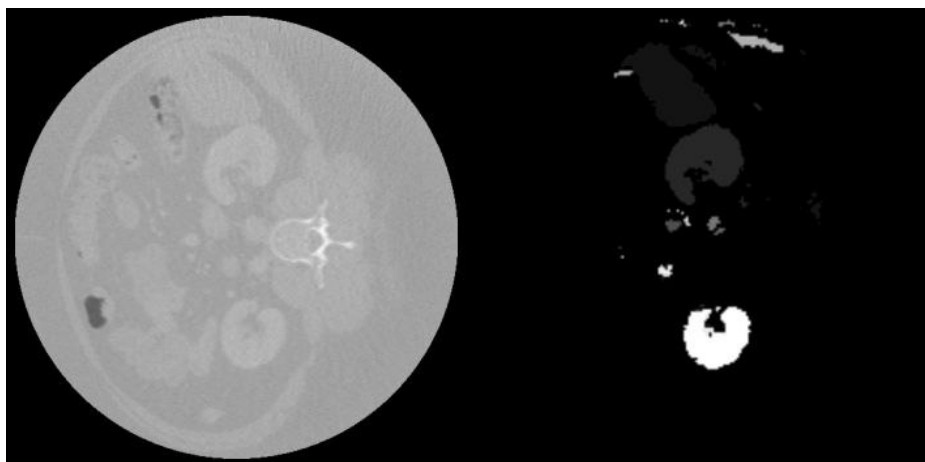

**Fig. 10.** Artifacts on predictions