# OpenReview forum: "CLEF: Contrastive Learning of Equivariant Features in CT Images"
_MICCAI.org/2022/Challenge/FLARE_

### Official Review · Reviewer_A5js · 2022-09-13
**clear description of the method**

**Rating:** 7
**Confidence:** 5

**Review:**

1. The description of the method is clear.
2. The images of the segmentation results are convinced.

---

### Official Review · Reviewer_j4r4 · 2022-09-17
**Great pretrained work about contrastive learning**

**Rating:** 9
**Confidence:** 4

**Review:**

Advantage:

1. It is very interesting that the pretrained method could work by contrastive learning, for the multi-organ's segmentation.

Disadvantage:

1.  Really look forward to more results before and after pretrained.
2.  More details of the pre-training process would be helpful and heuristics.

---

### Official Review · Reviewer_JZaV · 2022-09-17
**need to include ablation study**

**Rating:** 6
**Confidence:** 4

**Review:**

In Abstract, you can include your quantitative results and how much improved comparing with fully-supervised baseline. In the paper, you modified the 3D-Unet, it's good to mentioned some related works on modifications on 3D-Unet (if can please add related work).
Ablation study for modified 3D-Unet, since you modified the network, you want to find out the network performance when remove certain components and to evaluate contribution of the components.

"we manage to improve quality in comparison to a fully-supervised baseline" (from abstract). In 4 Results and discussion, mentioned it again. However, there no prove to it. That's being said, to draw a conclusion or a fact, you need to prove it. You can citate other's works that has this conclusion or run experiments to prove.

---

> ### Author Response · Authors · 2022-10-14
> **Thank you for your comments!**
>
> - The revised manuscript shall contain information on the baseline performance, to which we comapred ourselves, we thank you for pointing out this major flaw.
> - We shall also include our quantitative results in the abstract, along with the comparison to the baseline.
> - In essence, our version of the 3D-UNet is simply a 2D-UNet with 3D convolutions, 3D max-pooling and 3D interpolation. We don't consider this a major change from the basic 2D UNet architecture, which we cite in our work.

---

### Official Review · Reviewer_t9HH · 2022-09-22
**CLEF: Contrastive Learning of Equivariant Features in CT Images**

**Rating:** 4
**Confidence:** 3

**Review:**

Pros:
- Describe the pre/post-processing and the metrics in detail
Cons:
- more cases and the data in table should be discussed
- The performance of the network could be better if training more epochs

---

> ### Author Response · Authors · 2022-10-14
> **Thank you for your feedback!**
>
> - The pre/post-processing is the least of our contribution, and it is thus quite simple, but nevertheless the next revision will describe it in greater detail.
> - Sadly, we don't yet have much insight into the errors of our method, further discussion than that which is already present in the paper is beyond us.
> - Thank you for your recommendation concerning the training time, we shall take it into consideration in our future work.

---

### Comment · Reviewer_WXyn · 2022-09-25
**To some extent the manuscript is incomplete and not in proper shape, it is suggested to modify the whole manuscript.**

1: The abstract is not fully depicting your whole method; please revisit your abstract section.

2: Likewise, the whole paper has many missing components and did not fully provide an overall overview of your proposed method in precise detail.

3: There is no overall mean DSC and NSD, so we could not be aware of what actual mean DSC and NSD your method had achieved.

4: It is recommended to revise your whole manuscript and modify it accordingly.

---

> ### Author Response · Authors · 2022-10-13
> **Thank you for your insights!**
>
> Thank you for pointing out our errors, mistakes and imperfections, we will definitely take your insights into account. The revised version of our manuscript shall include the metrics, achieved by our model on the testing set, and describe our method in better detail.

---

### Meta-Review · Program_Chairs · 2022-09-28

**Recommendation:** Major Revision
**Confidence:** 5

**Metareview:**

Please change 1,2,... to 1), 2) ... in sec 2.1.
Reviewers raise many concerns and suggestions. Please address all comments in the revised manuscript.